# Polyphenol Supplementation and Antioxidant Status in Athletes: A Narrative Review

**DOI:** 10.3390/nu15010158

**Published:** 2022-12-29

**Authors:** Aleksandra Bojarczuk, Magdalena Dzitkowska-Zabielska

**Affiliations:** Faculty of Physical Education, Gdansk University of Physical Education and Sport, 80-336 Gdansk, Poland

**Keywords:** dietary polyphenols, athletes, antioxidant status

## Abstract

Antioxidants in sports exercise training remain a debated research topic. Plant-derived polyphenol supplements are frequently used by athletes to reduce the negative effects of exercise-induced oxidative stress, accelerate the recovery of muscular function, and enhance performance. These processes can be efficiently modulated by antioxidant supplementation. The existing literature has failed to provide unequivocal evidence that dietary polyphenols should be promoted specifically among athletes. This narrative review summarizes the current knowledge regarding polyphenols’ bioavailability, their role in exercise-induced oxidative stress, antioxidant status, and supplementation strategies in athletes. Overall, we draw attention to the paucity of available evidence suggesting that most antioxidant substances are beneficial to athletes. Additional research is necessary to reveal more fully their impact on exercise-induced oxidative stress and athletes’ antioxidant status, as well as optimal dosing methods.

## 1. Introduction

According to the WHO, nutrition is crucial for health and development. Better nutrition is linked to improved health, stronger immunity, lower risk of non-communicable diseases, and longevity (https://www.who.int/health-topics/nutrition, accessed on 11 July 2022). Dietary supplements are different from conventional food. They are concentrated sources of vitamins, minerals, or other ingredients such as polyphenol extracts. They are available in various forms, including tablets, capsules, gummies, powders, drinks, and energy bars. Dietary supplements are sold in pharmacies or health and beauty retailers, which might mislead patients and/or customers into supposing they are forms of drugs or medicine. In addition, certain scientific challenges are encountered in supplement research due to the absence of consensus on terminology for the categorization and regulation of these products. Nevertheless, dietary polyphenol supplements are widely used by athletes due to their multiple biological effects. This review discusses polyphenols as a major area of interest within the field of sports due to their antioxidative properties [1] that are important for athletes because muscle exercise induces oxidative stress and promotes reactive oxygen species (ROS) production [2,3,4]. Oxidation-reduction homeostasis is a critical process for human health as it regulates a myriad of biological responses [5]. Thus, polyphenols are considered key signal molecules upon introduction into the human body [6]. Recent evidence shows that oral administration of polyphenols strengthens the body’s antioxidant defense system and protects against oxidative stress in athletes [7,8]. However, the debate about polyphenols has gained fresh prominence with many researchers reporting disappointing outcomes [9,10,11,12,13]. Discrepancies in results remain, and the extent of the effects of polyphenols remains unclear [14]. The following review focuses on the use of polyphenols by athletes and aims to summarize current knowledge of the effects of dietary polyphenol supplementation on oxidative stress and oxidative damage. This review also includes details of polyphenol doses and exercise protocols.

### 1.1. What Are Polyphenols?

The term polyphenol is not well-defined. Surprisingly, the term was not found in the International Union of Pure and Applied Chemistry (UPAC) online databases nor in its glossaries [15,16]. Polyphenols are secondary metabolites of plants, meaning that they are synthesized through secondary metabolism and have no direct function in essential processes, i.e., photosynthesis or respiration [17]. However, they are involved in signal transduction from the root to the shoot [18]. For example, exogenously administered caffeic acid inhibits growth and enhances the lignification of soybean roots [19]. This indicates an allelopathic and allelochemical function of the acid. Allelochemicals released into the bulk soil might be absorbed by receptor plants [20], and can affect several plant processes, e.g., water utilization [21] or seed germination [22]. Interestingly, *Rhizobium* bacteria induce the synthesis of phenolic compounds in rice plants and rhizobial strains promote the growth and productivity of rice plants. highlighting a plant–microbe interaction [23]. Moreover, polyphenols are involved in the nutrient mobilization, e.g., by reducing soluble nitrogen in the soil [24]. The aforementioned examples outline the critical role of polyphenols in interactions with the general environment. 

It is worth noting that these secondary metabolites are natural products, and they occur naturally in, e.g., fruits, vegetables, cereals, and beverages [25,26,27]. They are also produced during abiotic stresses [18,28] through the phenylpropanoid pathway [29,30]. One potential example is salt stress, upon which a plant’s total phenolic content increases [31]. Salinity promotes ROS formation in plants, and this stress is mitigated by the plant’s antioxidative system, which relies on polyphenols [32,33,34,35]. 

Phenolic compounds have powerful antioxidant properties and are helpful in scavenging harmful ROS in plants under salt stress [36,37]. From a chemical point of view, they are characterized by phenyl rings and hydroxyl substituents [38]. However, the literature is not consistent in terms of polyphenols’ nomenclature and classification. Some publications describe two classes (flavonoids and non-flavonoids) [38,39], others three (phenolic acids, flavonoids, and non-flavonoids) [26,40], or four (phenolic acids, lignans, stilbenes, and flavonoids) [41,42,43]. Lignans (e.g., galbacin) are plant products of low molecular weight that consist of two propyl-benzene units coupled by β,β′ bonds [44]. Phenolic acids (caffeic, gallic and coumaric acids) generally have one carboxylic acid group [45]. Flavonoids have a basic structural unit of 2-phenylchromone [15]. Stilbenes contain a 1,2-diphenylethylene backbone [46]. The schematic classification of polyphenols is presented in Figure 1.

### 1.2. Importance of Research on Polyphenol-Based Dietary Supplements 

The global dietary supplements market increases in size every year. The market is driven by the rising popularity of sports-based and athletic performance enhancement supplements. While all regulatory scientists want to protect consumers from unsafe products, regulatory frameworks vary from country to country. A product might be considered a dietary supplement in one country, but prescription medicine in another. This means that pre-market research on polyphenol supplements is not mandatory worldwide. Scientific and regulatory challenges encountered in the existing research on the safety, quality, and efficacy of dietary supplements have been expertly reviewed elsewhere [49]. According to the National Institutes of Health (NIH), however, another problem lies in the fact that much of the research has focused only on single ingredients. Since dietary supplements for exercise and athletic performance contain multiple ingredients, a continued lack of data on the effectiveness and safety of the combinations in these multi-ingredient products is of high concern. A further complication is that most studies investigating the potential value and safety of supplements suggested to increase athletic performance involve mostly well-trained athletes. Therefore, there has been little discussion about their value to recreational or occasional exercisers. Moreover, many studies have involved young adults (more often males than females), and not adolescents who might want to use these dietary supplements (https://ods.od.nih.gov/factsheets/ExerciseAndAthleticPerformance-HealthProfessional/#h2, accessed on 11 July 2022). A further consideration relates to amounts of supplements in products. Some items provide neither the amount of each ingredient nor their standardization for, e.g., content of --polyphenols and anthocyanins. Only standardized extracts guarantee reliable clinical trials [50,51]. However, this practice is rarely implemented worldwide. In addition, polyphenols are of low bioavailability [47], which leads to poor therapeutic effects and will be discussed separately below.

A large and growing body of literature has investigated polyphenol-based supplements. Some of the emerging issues relate specifically to the experimental models used. Cell lines are frequently used, e.g., L929 mouse fibroblasts [52], lung epithelial carcinoma cell line (A-549), human ovarian cancer cell line (A-2780), human breast cancer cell line (MCF-7), human colon carcinoma cell line (Caco-2) [53], or RAW 264.7 macrophages [54]. However, one could argue that in vivo models seem more convincing than in vitro because they involve complex living systems, encompassing the process, pathway, or function under study while part of a larger “living test tube” [55]. By contrast, data from animal studies have been criticized due to the physiological differences between animal and human metabolisms [56]. Still, a considerable volume of in vivo polyphenol research has been reported. Examples of studies on human objects are elegantly reviewed in [47]. In animal models, mammals are used most frequently because they are biologically very similar to humans in evolutionary terms [57]. However, first and foremost, data derived from athletes is required because there is currently insufficient experimental evidence to convincingly suggest intake of preparations based on polyphenol antioxidants.

## 2. Properties of Polyphenol-Based Dietary Supplements 

Athletes experience physiological stress, which affects their whole performance. Effects include transient inflammation, oxidative stress, and immune dysregulation, which have been linked to nutritional influences [58]. The biological properties of polyphenols include antioxidant, anti-inflammatory, anti-cancer, anti-obesity, and anti-neurodegenerative effects, among more [59]. This review mainly describes the role of polyphenols in oxidative stress and inflammation, as these processes are linked. 

### 2.1. Polyphenols, Oxidative Stress, and Inflammation

As mentioned above, polyphenols are antioxidants. This activity is ascribable to hydroxyl groups operating as electron donors and stabilizing free radicals through the delocalization of unpaired electrons [60], and is termed a chain-breaking function [61]. Polyphenols also demonstrate metal chelation activity. Because transition metals can produce ROS that damage the genome, polyphenols play an antioxidative and DNA-protective role [62]. In addition, they inhibit certain enzymes involved in ROS production, i.e., xanthine oxidase and NADPH oxidase, whilst upregulating other endogenous antioxidant enzymes, e.g., superoxide dismutase (SOD), catalase, and glutathione peroxidase (GPX) [63]. The antioxidative property of polyphenols is important for athletes, because it is well established that exercise generates ROS formation and prolonged exercise can promote oxidative damage to active myofibers [64,65,66]. Oxidative stress presents an imbalance between ROS-generating and -scavenging systems. ROS contain at least one oxygen atom and one or more unpaired electrons. ROS include free radicals (e.g., superoxide anion (O_2_^−^), hydroxyl radical ·OH) and non-radicals (e.g., singlet oxygen (^1^O_2_), hydrogen peroxide (H_2_O_2_)). ROS overproduction can lead to tissue injury and initiate inflammation [63]. Superoxide (O_2_^−^) is relatively membrane-impermeable and unreactive [67] and has a short half-life in the cell [68]. Superoxide dismutase (SOD) spontaneously converts O_2_^−^ to non-radical hydrogen peroxide (H_2_O_2_) [69] which is further reduced to water by GPX [70]. H_2_O_2_ molecule is considered stable and can therefore diffuse considerable distances in or out of the cell [2]. O_2_^−^ and H_2_O_2_ are products of NADPH oxidase [71]. In contrast to H_2_O_2,_ the hydroxyl radical (·OH) is a strong oxidizing agent and is considered the most damaging [72,73]. Another free radical is nitric oxide (NO·), which is a short-lived and weak oxidant produced by the action of nitric oxide synthase (NOS) [74]. It has been shown that oxidative stress and inflammation are related, because inflammatory response occurs after exercise [75]. During inflammation, phagocytes undergo a respiratory burst that imposes high oxidative stress on the engulfed pathogens [76]. The oxidative burst is mediated by NADPH oxidase, which produces O_2_- and H_2_O_2_ [77], and these products achieve their cytotoxic effects due to reactions with other antimicrobial systems to generate ROS [67,78]. In skeletal muscle cells in culture, two primary NADPH oxidase isoforms (NOX2 and NOX4) [79] were found to be the main sources of ROS [80,81]. Thus, the role of NADPH oxidase-derived ROS is important in pathogen-mediated and sterile inflammation (i.e., without pathogen involvement). Importantly, ROS can exacerbate the inflammatory process [82]. Nonetheless, free radicals are crucial for proper cell physiology and function actively as intracellular messengers [68,82]. This means that ROS have both positive and negative physiological effects. ROS are by-products of cellular metabolism, and constitutive antioxidant defenses neutralize the housekeeping production of reactive species, e.g., superoxide dismutase (SOD) converts O_2−_ to H_2_O_2_ [83]. Another antioxidant strategy is the scavenging of ROS by numerous molecules of low molecular weight that exist in the extracellular space and also within cells. A further method is to minimize the availability of pro-oxidants via metal-binding proteins. These chelating molecules avert these transition metals from participation in ROS formation [2,84]. The toxic effects of ROS and RNS are also counteracted by non-enzymatic reactions, e.g., ascorbic acid or vitamin E [85]. However, when the production of radicals overwhelms the antioxidant defenses, oxidative stress proceeds. This can lead to changes in cell membranes and other biological structures including lipoproteins, lipids, proteins, DNA [86], etc. Polyphenols are of particular interest because they can also protect against oxidative damage through various mechanisms. For instance, resveratrol, which is a stilbenoid and a type of natural phenol found in red grapes, reduced the expression of NADPH oxidase 4 (NOX4) in mice [87]. Quercetin, a natural flavonoid found in onions, green tea, and apples, decreases lipid peroxidation and inhibits cellular oxidation in erythrocytes in type 2 diabetic patients [88]. Another flavonoid, myricetin also protected healthy human erythrocytes from oxidative stress in vitro [89]. The same effect was observed for tea catechins in human diabetic erythrocytes [90]. Conversely, others showed that catechins provided only modest protection from oxidative damage in healthy men [91] and showed no effect in soccer players [12]. Notwithstanding their high training demands, athletes’ diets may not contain sufficient antioxidants to support their physical activity [92,93]. This raises the question of whether they should take dietary polyphenols. Given that the health benefits of exercise are well documented, it seems unlikely that exercise-generated free radicals should negatively impact performance in the long term [92,94].

### 2.2. Polyphenols Bioavailability

The bioavailability of a substance is the fractional extent to which the active moiety is absorbed and rate at which the dosage reaches the therapeutic site of action [95,96]. The issue of bioavailability presents a major challenge for clarifying the therapeutic effects of polyphenols and may account for large inter-individual variations in clinical trials [97]. Many factors influence the bioavailability of dietary polyphenols in humans including external factors (sun exposure, degree of ripeness of plants), factors related to food processing (homogenization, lyophilization), interactions with other compounds (bonds with proteins (e.g., albumin), polyphenol-related factors (i.e., chemical structure, concentration in food)) [98,99].

The in vitro approach to studying polyphenols’ absorption, metabolism, and bioavailability has mostly used the Caco-2 cell line [100,101,102,103], because the line reliably reflects biological barriers and the ability to perform phase II biotransformation by adding hydrophilic groups. Nonetheless, other cells have also been used [104]. The in vivo approach mostly uses rodents [105,106,107,108] and humans [109,110,111,112]. Absorption of polyphenols in mammals occurs in the small intestine after cleaving off sugar moieties [113]. This process occurs by either passive diffusion or with the use of specific transporters localized in the enterocyte membrane [99]. Only approximately 5–10% is absorbed, and the remainder accumulates in the large intestine with the bile conjugates, being thereby subjected to digestion [114]. Importantly, polyphenols of low molecular weight such as gallic or caffeic acids are partially absorbed into the body directly or after conversion. In contrast, polyphenols of higher molecular weight, e.g., proanthocyanidins, are very poorly absorbed [115]. Phenolic compounds absorbed in the small intestine are further transported to the colon, where they are transformed by the gut microbiota. These microorganisms perform a myriad of activities including hydrolysis, dihydroxylation, demethylation, and decarboxylation [99] thereby converting phenolic compounds into bioavailable metabolites. Meanwhile, polyphenol-rich extracts can also modulate gut microflora [116,117]. Importantly, the position of the hydroxyl groups might influence the polyphenols’ breakdown. For instance, flavonoids without hydroxyl groups at the C5, C7, and C4′ positions are degraded more slowly [113,118]. This implies that the slow-degrading compounds might be more bioavailable because their absorption can occur more quickly than those degraded more rapidly at the colon level [119]. When polyphenol sub-products are absorbed by the small intestine or colon, phase I and II transformation begins within enterocytes and in the liver to be distributed through the blood [120,121,122]. However, the metabolites may encounter enterohepatic circulation, whereby they are returned to the small intestine and transported back to the liver [97,99,123]. Polyphenols can stay in the blood or be delivered to other tissues. Indeed, polyphenol metabolites have been found in various tissues, e.g., glucuronidated and methyl-glucuronidated derivatives of catechin and epicatechin in the muscle of rats fed with grape pomace extract [124] or catechin-glucuronide, methyl catechin-glucuronide and methyl catechin-sulfate in kidneys, intestine, lungs, spleen, and thymus of rats receiving hazelnut extract [125]. Nonetheless, their targeted delivery is a major problem relating to polyphenol-rich supplements and their low bioavailability. So far, many systems have been developed to address the issue. However, these approaches mainly concentrate on achieving delivery that minimizes systemic diffusion and degradation of phenolic compounds, because some phenolics are poorly absorbed in the digestive tract while others are extensively metabolized to derivatives with a lower activity or are degraded [99,126]. This means that when polyphenols are orally administered, their therapeutic concentration might not be achieved [127]. Temperature, light, oxygen, acidic pH, and enzymatic activity in the digestive system [127] can all affect the beneficial effects of polyphenols [128]. As mentioned above, polyphenols follow phase I and II metabolic pathways. The major pathway is phase II, during which glucuronidation and sulfation lead to especially hydrophilic conjugates, and methylation to similar if not slightly more lipophilic metabolites. The methylated products are conjugated into glucuronides and sulfates, as long as there is a functional group available for conjugation into more hydrophilic metabolites [120]. Moreover, unabsorbed and hydrophilic polyphenol conjugates can undergo transformation triggered by gut microflora, which can even result in ring fission. Bacteria can dramatically reduce polyphenols’ bioactivity [129]. Therefore, various research efforts have been focused on the problem of polyphenol delivery. Of note, bioavailability is a key determinant of polyphenols’ potential health-promoting applications [99], and is also important in establishing dietary reference intake (RDI) [130]. This aspect is reviewed in the discussion of polyphenol supplementation strategies. 

### 2.3. The Intake of Phenolic Compounds in an Average Daily Diet

The biodiversity of polyphenols found in food is wide. Therefore, it is extremely difficult to determine their content within food products and measure their daily intake [131]. Several research articles have extrapolated the daily dietary intake of particular flavonoids among the European population, based on the US database of flavonoid concentrations in particular foods. Many authors have referred to the data published by Kühnau et al., almost 46 years ago, where a daily intake of 1 g of total phenols was established in the US population [132]. There have been attempts to define the daily consumption of polyphenols within the diets of different populations. So far, Hertog et al., have done this for flavonols and flavones [133], and Reinli and Block for isoflavones in the Dutch population [134]. Researchers established the intake of flavonols (mainly quercetin) and flavones as 21 and 2 mg/d, respectively [133]. The average dietary consumption of isoflavones in the Japanese population was determined to be 30–40 mg/d [135,136]. Interestingly, soy products are less popular in Western countries, which is reflected in the lower polyphenol content of the Western diet [137], explaining why the daily dietary intake of quercetin and genistein does not exceed 2–4% of total polyphenols [131]. Some authors report that daily polyphenol consumption above 650 mg decreases risk of death in comparison with those whose daily polyphenols intake is below 500 mg [138]. Other authors report health benefits ranging from intakes of 500 mg to 1500 mg per day [131]. Further sources recommend a daily dose of 0.1–1.0 g of polyphenols. Fruits such as grapes, apples, pears, cherries, and blueberries contain up to 200–300 mg of polyphenols per 100 g of fresh weight. Interestingly, a glass of red wine or a cup of tea contains about 100 mg of polyphenols, and the presence of these products in the diet may reduce the likelihood of chronic diseases. In Europe, the main sources of polyphenols are onions, black tea, red wine, and apples [138,139].

### 2.4. Polyphenols Delivery

Various delivery systems for polyphenols have been developed to improve their efficiency and combat their bioactivity problems. One major issue is to find a way to enhance the penetration of active substances and bring hydrophilic compounds into the tissues. This concept has been explored in numerous studies demonstrating the use of formulations with simple emulsions [140,141,142], cyclic glucan oligosaccharides known as cyclodextrins [143,144,145], gels [146,147,148], nanoemulsion [149,150,151], or liposomes [152,153,154]. Other solutions are also available, such as micelles, nanocomposites, metal oxide nanoparticles, etc. [127]. The main reason for encapsulating polyphenols is to tackle the problem of their stability. For instance, maltodextrins preserve the integrity of anthocyanins [155]. Encapsulation increases biocompatibility and prevents degradation caused by the external environment. It also minimizes interactions with other components of the human body [156]. In general, coating by microencapsulation in particles up to 1000 μm protects active substances and preserves their antioxidant properties [157,158,159]. However, nanotechnology is apparently more effective, because nanoencapsulated polyphenols have been found to increase the protection of active substances and bioavailability as well as improving controlled targeted release [160]. Particle size is generally seen as a factor strongly related to bioavailability. The development of a nanoscale delivery system is aimed at achieving improved site-selective targeting [161]. This is possible due to the small molecular size and active incorporation into cells by different endocytic pathways [162,163]. However, targeted delivery is complicated. It can be achieved actively or passively [164]; active targeting involves the therapeutic agent (in this case a polyphenol) being loaded into a carrier and this conjugate attached to tissue or cell-specific ligands [165]. By contrast, passive targeting requires loading the therapeutic agent into a nanomolecule that passively reaches the target tissue or organ. This leads to the accumulation of a drug delivery system with a specific size, molecular mass, and charge [166]. Given that a passive targeting system might utilize specific conditions in the diseased tissues or cells (e.g., low pH) [166], it is reasonable to assume that this approach is suitable to treat oxidative stress generated by exercise. After all, the higher the exercise intensity, the lower the muscle pH [167]. Nonetheless, the active approach also seems effective since it is possible to target specific cells or the inside of cells (for targeting intracellular organelles) [166]. 

Importantly, approaches should be compatible with another target, that of achieving the concentrations required for systemic therapies. In these circumstances, the hydrophilic part of a phenol carrier must be in balance with the lipophilic part [168], because the affinity of polyphenols for lipid bilayers partially determines their biological activity in vitro [169] and is of great significance in the biomedical and dietary fields. Polyphenolic extracts interact with the cell membrane by creating a protective coat around the lipid membrane, through their location on the membrane surface. This effect was shown using liposomes as models of lipid membranes, wherein trans-stilbenes and flavonoids interacted at the hydrophilic interface [170]. This finding is in agreement with another study reporting that three different types of blueberry extracts changed the arrangement of the hydrophilic region of the liposome membranes [171]. Liposomes themselves possess an aqueous central section as well as hydrophobic and hydrophilic components comprising a lipid bilayer. The aqueous cores typically encapsulate hydrophilic compounds. By contrast, hydrophobic substances favor lipid bilayers. Thus, liposomes might be used for the delivery of diverse substances, such as hydrophilic and hydrophobic compounds [172]. 

## 3. Oxidative Stress and Athletic Antioxidant Status 

### 3.1. ROS in Working Muscles

Contracting muscles produce ROS, as shown in muscle tissue homogenates and intact muscles in rats [65]. While cellular respiration generates ROS, there is no hard evidence that mitochondria are the main source of ROS in contracting muscle fibers [173]. Early findings indicate that superoxide production in isolated rat or bovine mitochondria occurs at a level of 2–5% of total oxygen consumed by mitochondria [174,175]. Interestingly, recent results differ from that 2–5% estimate of superoxide generation; it was reported that less than 0.15% of oxygen consumed by the mitochondria is used in superoxide [176]. This was demonstrated using intact isolated rat mitochondria. These findings may disprove the proposal that mitochondria release significant amounts of reactive oxygen species under physiological conditions. In addition, in rat skeletal muscle fibers produce more ROS in the basal state of respiration compared with the active state [177]. It is therefore questionable whether mitochondria are the primary source of ROS production in the working muscle. Another debate concerns whether exercise-induced ROS generation is detrimental to health. This is somewhat doubtful, because if it were the case then people who engage regularly in sports would experience a higher prevalence of chronic diseases associated with oxidative stress. In support of this, the study by Walsh et al., demonstrated that endurance training increased VO_2_max by 21% and did not increase the sensitivity of mitochondrial oxidative function to ROS in skinned fibers [178]. By contrast, sprinting relies mostly on anaerobic energy pathways [179] and high-intensity sprint training inhibits mitochondrial respiration [180]. Thus, exercise can be perceived as a double-edged sword, whereby moderate ROS production initiates positive physiological adaptation in the active skeletal muscles, and meanwhile high levels of ROS lead to damage in, e.g., proteins, lipids, and DNA [2]. Previous studies on high-level swimmers have reported that high-intensity discontinuous training and continuous moderate-intensity training produced similar levels of oxidative stress expressed by plasma biomarkers such as protein carbonyl, thiobarbituric acid-reactive substances, and 8-hydroxy-2-deoxy guanosine. The obtained results suggested the beneficial role of exercise independent of the intensity of training [181]. Likewise, although high-intensity discontinuous training induces ROS production, a gradual decrease of ROS and return to resting values is typically seen after 10 min. Adaptive responses are responsible for this phenomenon. The human body can expand its antioxidant capacity to restrain oxidative chain reactions and thereby control oxidative stress during exercise [182]. However, reactive species appear to act in a hormetic manner [181,183]. For example, subjects with low total antioxidant capacity (TAC) values prior to training showed an increase in these after exercise. Conversely, participants with high pre-training values showed a decrease [181]. It therefore seems likely that individual differences might occur in response to training. 

Furthermore, the source of ROS during anaerobic exercise is unclear. Ex vivo muscle fiber data from mice shows that NADPH oxidase is one of the potential sites of superoxide, both at rest and during contractions in skeletal muscle [184]. Nevertheless, the measurement of ROS release by muscle mitochondria isolated after exercise from humans [185] and animals [186] suggests that aerobic exercise increases ROS production. It has been even suggested that a source other than the mitochondria initially releases the ROS, damaging the mitochondria and leading to their elevated ROS release [187]. Nonetheless, exercise leads to microinjury in muscle tissue [188,189], leading to increased tumor necrosis factor alpha (TNFα) levels due to increased TNF expression by damaged muscle fibers in addition to macrophage infiltration [190,191,192]. Macrophages respond to changes in their cytokine microenvironment; briefly, they undergo polarization to either the M1 or M2 activation phenotype following exposure to a specific stimulus [193]. Activated macrophages produce reactive oxygen species [194]. TNFα has been shown to interfere with oxidative phosphorylation in isolated rat and bovine mitochondria [195] and to stimulate ROS production [196]. Strenuous exercise also increases levels of serum platelet-derived growth factor (PDGF) [197], resulting in rapid ROS generation [198]. Plasma levels of Interleukin-6 (IL-6) depend on the intensity and duration of exercise, and IL-6 is sharply reduced when the muscle becomes more energetically efficient [199]. However, a pleiotropic Interleukin-6 (IL-6) dysregulates the muscle redox balance, as shown ex vivo in mice [200,201], but eventually reduces ROS production, which was demonstrated in vitro in the mouse myoblast cell line [202], whereas chronically elevated IL-6 was shown to contribute to oxidative stress in isolated mouse muscles [200,203]. Human ex vivo data indicate that the increase in pro-inflammatory cytokines is balanced by the release of anti-inflammatory cytokines such as Interleukin 1 receptor antagonist (IL-1Ra), IL-4 [204], and IL-10 [205]. A shift from pro- to anti-inflammatory signaling after muscle injury enables the termination of the inflammatory response and supports later phases of myogenesis, which was shown ex vivo in isolated mouse muscles [206]. This process is highly complex, as not only muscles and macrophages but many other cell types and signaling molecules are involved. Ultimately, rebalancing between pro- and anti-inflammatory cytokines prevents excessive ROS accumulation and guarantees full muscle rebuilding and return to redox homeostasis [207]. 

Interestingly, the appearance of ROS in damaged human-muscle fibres fosters muscle remodelling and their adaptation to physical activity [208,209] for two reasons: (1) ROS cooperates in activation of transcription factors, e.g., nuclear factor kappa-light-chain-enhancer of activated B cells (NF-κB) [210], which regulates expression of several genes including cytokine genes. NF-κB activation stimulated by contracting muscles provides a foundation for adaptation to exercise and regulates the expression levels of antioxidant enzymes in human skeletal muscle [211]; (2) ROS activate proteasomes [212] that degrade damaged proteins sensitive to ROS, i.e., actin. This eventually increases proteolysis and synthesis of proteins and intensifies the anabolic effect. In summary, ROS have divergent roles in humans. It should be highlighted that only a fully functioning antioxidant defense system maintains ROS at a safe level [213]. Normally they are generated at low levels and restricted to a specific subcellular location, thereby initiating a multitude of signaling cascades supporting normal physiological processes [214]. A schematic illustration of the possible representative pathways of polyphenol action in the muscle is depicted in Figure 2, with the example of lignans. Briefly, lignans decrease protein degradation, enhance protein synthesis, reduce oxidative stress, and improve mitochondrial biogenesis and myogenesis [215].

### 3.2. Physical Activity, Oxidative Stress and Antioxidant Status 

Oxidative stress markers are molecules modified by interactions with ROS in the microenvironment. Quantifications can be classified into ROS detection, measurements of antioxidant levels (e.g., glutathione peroxidase (GPX), superoxide dismutase (SOD), catalase (CAT)), oxidation products (e.g., protein carbonyl (PC), thiobarbituric acid-reactive substance (TBARS) markers of protein oxidation, F_2_-isoprostanes, and malondialdehyde (MDA), which are markers of lipid peroxidation, 8-oxo-2′-deoxyguanosine (8-OHdG), which is a marker of deoxyribonucleic acid (DNA) oxidation), or measurements of the redox balance (e.g., the reduced glutathione/oxidized glutathione (GSH/GSSG) ratio) [94]. Oxidative stress can be assayed by total antioxidant status (TAS) and the free radical scavenging capacities of plant extracts can be measured with methods such as 2,2-Diphenyl-1-picrylhydrazyl (DPPH), 2,2′-azino-bis(3-ethylbenzothiazoline-6-sulfonic acid) (ABTS), oxygen radical absorbance capacity (ORAC), total antioxidant capacity (TAC), etc.

Numerous publications highlight that oxidative stress and ROS generation sharply increase after the workout. For example, a short-term training period (repeated sprint) performed by elite soccer players led to elevations in TBARS, PC, SOD, and CAT [216]. Another study demonstrated that ultra-endurance running increases PC, TBARS, TAC, and 8-OH-dG immediately after testing in trained endurance athletes [217], whereas others reported no changes in TBARS, PC, and TAC observed during or after running [218]. Lower plasma malondialdehyde (MDA) was found in trained subjects compared to untrained, at rest 15 min after strenuous exercise [219]. However, contradictory findings have also been reported. Handball players undergoing 6 months of training were characterized by lower MDA results compared with non-athletes [220]. Similarly, treadmill runners maintained MDA at a stable levels over a 9-month program [221]. This suggests adaptation to regular exercise. Furthermore, changes in oxidative stress and redox status according to training load have been described. For instance, PC and TBARS were elevated in soccer players at the end of the pre-season training period, reduced in the middle of the season, and increased again at the end of the season. In contrast, antioxidant status biomarkers showed an opposite pattern of variations [222]. In another example, significant elevations of TBARS, PC, CAT activity, TAC, and also significant decreases in the GSH/GSSG ratio were found [223]. The ratio GSH/GSSG represents whole-body redox status [224] and its reduction is indicative of oxidative stress, as reported previously [225]. These results were observed in boys and girls participating in intense swimming training [223], and are in line with others who reported that antioxidant status and oxidative stress develop gradually over a sports season [226] and the redox status of professional athletes is influenced by training load throughout the season [227]. Although athletes’ performances demonstrate various responses expressed as redox homeostasis markers, the data imply that short-term and long-term exercise are obvious stressors for the athlete’s redox status, resulting in acute and chronic responses, respectively [228]. Moreover, oxidative stress and antioxidant status might relate to training intensity, as supported by the work of Lamprecht et al. This group examined regular aerobic exercisers and investigated the effects of single bouts of exercise at three different intensities on the redox state of human serum albumin, which acts as transport and redox system, and proposed that the redox pool was accessed more completely with increasing exercise intensity [229]. Furthermore, antioxidant status seems to relate to training status. In a study comparing untrained versus trained participants, the former showed (a) a larger increase in PC in response to exercise at all times throughout the training, (b) a more pronounced response from TBARS immediately and 1 h after training, (c) lower TAC values at rest and in response to exercise throughout the test period, and (d) lower GSH values at rest during the test period, and immediately and 1 h after exercise [230]. In addition, the literature indicates that the type of exercise might exert a different effect on the level of ROS-induced end products [3,231].

Lastly, antioxidant levels in the blood increase during physical exercise [232]. It also appears that these levels depend on the types of sport involved. Interestingly, as SOD activity increased, GPX and CAT activity also increased, and this finding was associated with the physical condition of interval-trained athletes [233]. Changes in GPX, SOD, and GR confer a counteraction to radical production [231]. Ex vivo experiments suggest that regular training might increase the activity of antioxidant enzymes and reduce oxidant production [220,234]. 

Taken together, these results suggest antioxidant status is of great importance in terms of health maintenance. Despite a large volume of studies describing the negative aspects of exercise (ROS and oxidative stress), supportive data also exists and many studies continue to demonstrate the health benefits of exercise. These benefits include adaptation mechanisms, e.g., a rise in antioxidant capacity with increasing training effort as shown ex vivo in adolescent athletes [235]. Thus, it seems unlikely that oxidant production via exercise negatively impacts performance and health in the long term [92]. Although differences of opinion exist, it is widely accepted that exercise-induced radical production can damage skeletal muscle fibers and lead to fatigue. This knowledge has prompted many athletes to consume antioxidant supplements [93].

### 3.3. Physical Activity, Stress Hormones, Oxidative Stress and Antioxidant Status 

Physical and psychosocial stressors activate the sympathomedullary pathway (SAM) and the hypothalamic–pituitary–adrenal axis (HPA axis) [236], and these routes trigger release of stress hormones. For instance, the SAM pathway leads to catecholamine (e.g., adrenaline and noradrenaline) secretion from the adrenal medulla, which are part of a short fight-or-flight response. The HPA pathway induces a longer transient hormonal cascade leading to glucocorticoids (cortisol) release from the adrenal cortex [237]. Researchers have studied the close relationship between stress hormones and athletic performance [238,239], and it has been found that circulating catecholamine levels increase during exercise in healthy athletes [240]. Catecholamines have beneficial effects on the heart, skeletal muscle, and central nervous system in relation to exercise performance [241]. Nonetheless, during pronounced oxidative stress in sustained or ongoing exercise, excess secretion of catecholamines contributes to immune suppression and a rise in ROS production due to catecholamine peroxidation [242,243,244]. Catecholamines released during exercise can also bind to β-adrenergic receptors, increasing sympathetic activity and alleviating skeletal muscle oxidative metabolism [245]. 

Competitive situations also change the levels of cortisol, the primary hormone of the HPA system, axis activation of which improves alertness for extended durations [246]. The paraventricular nucleus (PVN) of the hypothalamus releases corticotropin-releasing hormone (CRH). Adrenocorticotropic hormone (ACTH) is then released from the anterior pituitary and acts on the adrenal cortex to release cortisol. According to Filaire et al., the typical cortisol response pattern exhibits an anticipatory spike before competition [247] and a second rise shortly after competition, as a result of extreme mental and physical strain [248]. The cortisol–performance relationship is inconsistent. For instance, there was no correlation between cortisol levels measured before multiple games of basketball and retrospective athlete and coach performance evaluations [249]. A negative correlation between performance and cortisol levels on the day of competition was observed during a 36-hole golf competition, suggesting that higher cortisol levels are linked to worse golfing performance [250]. In a study that utilized more time-points, cortisol levels in the loser of a tennis match were higher before (17%), during (65%), and after (54%) the game. Additionally, there was a negative correlation between cortisol and other performance indicators (e.g., unforced errors and return performance) [251], implying variations in cortisol. Another compound worth mentioning is 11β-hydroxysteroid dehydrogenase (11β-HSD), which catalyzes the conversion of cortisone to biologically active cortisol. According to Dovio et al. [252], strenuous exercise markedly elevates systemic 11b-HSD type 1 activity in humans. In addition, their findings suggest that 11b-HSD activity can profoundly upregulate cellular concentrations of cortisol. Therefore, it is plausible to suppose that increased local endogenous glucocorticoid availability might help to reduce exercise-induced muscle damage by helping to stop the inflammatory response in the muscles [252]. Meanwhile, when cortisol levels are elevated, glucocorticoid-induced muscle atrophy might occur with the breakdown of contractile proteins. It was also reported that exercise-induced cortisol is linked with reactive oxygen species production [253]. Interestingly, antioxidant supplementation has been linked with reduced secretion of cortisol in athletes [254,255,256].

### 3.4. Dietary Polyphenols and Protocol Considerations

Oxidative stress is an active field of athletic research, and a wide range of different oxidative stress biomarkers has been used for its assessment. Since these markers typically result from diverse reactions, they also vary in how quickly they manifest and vanish following exposure [257]. Furthermore, oxidative stress indices also have drawbacks. For instance, in the TBARS assay in vitro and/or ex vivo, up to 98% of the measured MDA might be produced by the high temperature during the protocol itself [258]. TAC measures the total antioxidant capacity of plasma-bound small molecules, excluding peroxidation and the crucial role of intracellular enzymes including SOD or CAT. Thus, measurements cannot be taken as an accurate representation of in vivo antioxidant capacity [259]. An additional direct measurement technique for assessing ROS generation ex vivo is electron paramagnetic resonance [182,260], while the majority of studies use indirect markers of oxidative stress as surrogate markers [261]. These factors along with the pharmacokinetics of different dietary supplements must be considered when planning the blood-collection schedule of a study [257]. Thus, although a placebo-controlled crossover study design seems ideal and straightforward, many complex choices must be made, e.g., the length of pre-supplementation, timing of intake before testing, the dosage of the supplement, and the ideal washout period [257]. The gold standard for studying the effects of dietary supplements on athletic health should also include verification that the dietary supplement was ingested and elicited a biological response (e.g., by muscle, blood, urine, or saliva sampling) [262]. 

In elite athletes, it might be important to take into account the part of the season in which acute study was carried out. Overloading of exercise during the early part of the season might put additional stress on the body’s immunological, muscular, and antioxidant systems [257,263,264,265]. The literature also points to an inter-individual variability of redox responses [266]. According to Lehmann et al., individual heterogeneity in recovery potential, exercise capacity, stress tolerance, and training tolerance accounts for the different ways in which athletes are vulnerable to overtraining under the same training conditions [267]. Even in subjects with the same training status, variations between individuals regarding the changes in oxidative stress markers were found ex vivo [268]. Inter-individual variability in response to consumption of polyphenols has also been reported [265]. Firstly, the bioavailability of polyphenols is impacted by inter-individual variability of absorption, distribution, metabolism, and excretion. Secondly, the target tissues’ cellular molecular metabolism and the inter-individual variability of the cellular processes influenced by the bioavailable polyphenols (and/or their metabolites), are likely to have an impact on the bioactivity of these compounds [269]. A range of factors including genetic background, gut microbiota, age, gender, health, and training status may be involved in these inter-individual variations, but the current knowledge is sparse and fragmented. Another major problem in the literature investigating the effects of dietary polyphenols lies in the fact that many researchers did not use pure polyphenols or their mixtures, but blends with other antioxidants (e.g., juices containing polyphenols and vitamins [270,271,272,273,274]). Therefore, the antioxidant role of the active molecules could be ascribed not only to the polyphenols but also to, e.g., vitamins or the combined effect of polyphenols and vitamins, the latter also being found in berry juices. In other words, if there is no comprehensive information regarding the supplement content, i.e., if the supplement did not consist of polyphenols only, the experimental results cannot be attributed to polyphenols alone. This does not mean that the product to be tested cannot contain auxiliary substances or fillers. Nonetheless, the ideal dietary polyphenol supplement should be standardized to ensure the constant concentration of polyphenols, their quality, and the therapeutic efficacy of each product without batch-to-batch variations. In contrast, unstandardized plant extracts contain variable amounts of active substances and use of the same dose does not always produce the same therapeutic effect. For instance, Lafay et al., studied the performance of elite male athletes given supplements of 400 mg of grape extract. This extract contained no purified polyphenol, and chemical analysis showed that gallic acid, catechin, and epicatechin were present in comparatively high amounts. The authors stated that the extract was from whole grapes, which means that the seeds might have been used [266]; these contain vitamin E, which is an antioxidant [275,276]. Therefore, the findings that grape extract improves antioxidant status and physical performance in elite male athletes do not seem to be polyphenol-specific. Another example can be found in the work Morillas-Ruiz et al. [270], who provided professional cyclists with a supplement in the form of a sports drink containing berry concentrates (total ingested polyphenol 2.3 g), vitamin C (20 mg/L), maltodextrin, pectin, and whey protein (unspecified quantities), as well as vitamin B1 (15% of recommended daily intake). Vitamin C was present in the placebo, which contained but no carbohydrates or other active ingredients. The fruit concentrate of the supplement drink (black grape [81 g/L], raspberry [93 g/L], and red currant [39 g/L]) provided almost all of its carbohydrates. The concentration of anthocyanin was highest (759 mg/L), followed by hydroxycinnamic acid derivatives (246 mg/L) and ellagic acid (168 mg/L) [270]. The authors concluded that the supplement protected against oxidative stress [270]. However, because the dietary supplement contained berry concentrates that provided polyphenols, carbohydrates, and vitamin C it is difficult to attribute these differences to polyphenols alone. An equivocal conclusion was also found in the study by Pilaczynska-Szczesniak et al., who supplemented male rowers with chokeberry juice (23 mg of anthocyanins in 50 mL of juice, three times per day for four weeks). Ex vivo analysis demonstrated that following a 2000 m ergometer rowing test, TBARS were dramatically decreased 24 h post-exercise in the supplemented group as opposed to those receiving the placebo [277]. The authors did not assess vitamin C content, found in aronia fruit which is a good source of vitamin C. The latter is well known for its antioxidative properties [278]. Thus, since the supplement might potentially have provided vitamin C as well as polyphenols, it is difficult to assign the differences between the supplement and placebo groups to the polyphenols alone. 

As described, there are a number of methodological considerations regarding polyphenol supplementation in athletes, which can greatly affect experimental outcomes. Meanwhile, the optimal dosage and dietary reference intake remain undetermined. These issues are discussed in the supplementation strategies section, below (Section 3.5).

### 3.5. Polyphenols Antioxidant Intervention and Antioxidant Status in Athletes

Some athletes endorse dietary supplements based on data that supports improved exercise performance [279]. Studies have confirmed that polyphenols. i.e., quercetin [280], resveratrol [281], and polyphenolic compounds from grape extract [266] or beetroot juice [282,283,284] improve exercise performance. Quercetin was found to increase mitochondrial biogenesis and endurance capacity in mice ex vivo [285]. This was shown by increased mRNA expression of *PGC-1γ* and *SIRT1*, as well as increases in cytochrome c concentration [285]. *PGC-1γ* and *SIRT1* promote mitochondrial biogenesis [286]. Increased cytochrome c concentrations coincidence with other concentrations of mitochondrial enzymes involved in the electron transport chain (ETC), the tricarboxylic acid cycle (TCA), and the β-oxidation pathway, which increases mitochondrial capacity [285,287]. Mitochondrial biogenesis is important in sports performance because it expands the total mitochondrial volume and improves capacity to respond to contraction as cells increase their numbers of mitochondria [288]. Resveratrol, which has a similar structure to quercetin, also improves mitochondrial function and aerobic capacity in mice [281] and rats [289], as shown experimentally ex vivo. 

The reason for the interest in polyphenols’ antioxidants is the discovery that highly reactive chemical species can increase during exercise. Unfortunately, there are very limited and contradictory studies on the effects of polyphenol supplementation on exercise-induced oxidative stress and antioxidant status in athletes [290,291]. In contrast, there is abundant literature on exercise in animals [292,293,294,295,296] or non-athletes [271,272,297,298,299,300,301]. In humans, blood is the most frequently analyzed biological sample, while a limited number of studies have examined skeletal muscle. For instance, 8-week supplementation with 500 mg/d hesperidin, a flavanone abundant in citrus fruits [302], was shown to reduce oxidized glutathione in amateur cyclists, indicating reduced oxidative stress. The effect was not seen in the placebo group [303]. This is in line with in vitro studies where hesperidin was demonstrated to reduce the by-products of lipid peroxidation, measured as TBARS in human erythrocytes [304] and MDA in human hepatic cells [305]. TBARS also decreased in cyclists after one-time consumption of a polyphenol blend including black grape, raspberry, and redcurrant (2000 mg polyphenols, including 1212 mg anthocyanins) [270]. Furthermore, 1 month of supplementation with 400 mg grape extract in elite athletes resulted in decreased creatine phosphokinase (CPK), a marker of skeletal muscle damage, supporting the idea of improved recovery [266]. It was also shown that oral administration of grape extract increased antioxidant capacity, while levels of SOD, CAT, and lipid peroxidation markers did not change between the placebo and treatment groups. However, significant results were obtained for isoprostanes and athletic performance in handball players. Although not all markers demonstrated the anticipated results, the data were encouraging. The authors suggested that the grape extract might be beneficial for maintaining the balance of oxidative stress and antioxidant status balance, enhancing physical performance in one category of athlete [266].

In studies of catechins, which represent flavanols, a single dose of 640 mg of green tea catechins in soccer players did not impair exercise-induced oxidative stress or muscle damage [12]. Similarly, Dean et al. concluded that 6-day intake of 270 mg epigallocatechin-3-gallate by male cyclists had no effect on TBARS [306]. However, data for dark chocolate, which is flavonoid-rich [307], appears very convincing. Consumption of 80 g/day for 2 weeks resulted in reduced plasma F2-isoprostanes at exhaustion and after 1-h recovery compared with the placebo group [308]. That study produced results that corroborated the findings of Davison et al., who found that 100 g of dark chocolate 48 h before a cycling challenge significantly increased TAS [299]. Chronic intake of dark chocolate (40 g/day equivalent) was also able to reduce oxidative stress and muscle damage biomarkers during elite football players’ training sessions [309]. The opposite was seen for Montmorency cherry (600 mg of phenolic compounds and 40 mg of anthocyanins 10 days before testing) in resistance-trained males. Following resistance exercise, markers of oxidative stress, lipid peroxidation, and antioxidant activity did not change significantly over time [310].

The redox-related effects of supplementation with a wide range of phenolic compounds on oxidative stress induced by severe physical exercise was the subject of a 2022 review by Kruk et al. These effects were thoroughly reviewed but the research largely focused on combined effects of polyphenols in sports and not specifically on antioxidative parameters. However, they listed 12 publications in the recent literature that demonstrated the reduction of oxidative stress markers or improvements in antioxidant capacity in professional sportsmen [290], although the list included sedentary and other non-athletes that are beyond the scope of this current review. Therefore, based on [290] and our own research, relevant positions in the literature are denoted that directly relate to polyphenol intervention and redox status in athletes [11,12,270,271,272,308,311], summarized in Table 1. The table does not refer only to polyphenol intake alone. For instance, the research reported by Jówko et al., 2015 [11] used green tea extract, which might contain antioxidant compounds including flavonoids, epigallocatechin-3-gallate, and vitamins. Further examples can be found [266,310,312,313,314,315] wherein Montmorency cherry extract was examined. None of these publications except from Levers et al., 2015 [310], mentioned whether the skin of the Montmorency cherry was used for preparation of the extract. The skin contains melatonin, which acts as an antioxidant [316], but none of these publications analyzed the extract for melatonin. This means that the effects described in Table 1 cannot be ascribed to polyphenols alone. It is interesting to note that some of the included research reported significant drops in just a few oxidative stress markers while finding no change in others. In addition, there was variation in the doses used. Research has shown that selected antioxidant doses might not always be advantageous or effective. For instance, 4-day supplementation with 480 mg of resveratrol daily had no effect on oxidative stress when compared with a placebo in physically active men subjected to a cycling challenge [13]. Authors have emphasized the presence of a variety of antioxidants in human cells, some of which are enzymatic in character, such as CAT and SOD. Intracellular antioxidants often reach millimolar concentrations, whereas polyphenols’ circulating concentrations normally do not exceed the low micromolar range [317].

The maximal daily dose of dietary polyphenols remains unknown because they have no established dietary reference intake value (DRI) [59]. This is because polyphenols are classified as nutritional supplements and no such value exists in the pharmacopoeia. Classically, DRIs include the recommended dietary allowance (RDA), adequate intake (AI), the tolerable upper intake level (UL), and the estimated average requirement (EAR). All of these values are linked; no RDA will be set if an EAR cannot be established. When evidence remains insufficient to develop an RDA, AI can be established. The UL is not intended to indicate a recommended level of intake (RDA or AI) but warns against a harmful excess of nutrients [318]. The DRIs apply to the healthy general population, which implies a gap in the knowledge regarding ill individuals. In addition, it might be that life stage and the subject’s sex can affect DRIs. Nonetheless, these values for polyphenols have not been set. To complicate matters, there exist numerous polyphenols with diverse structural properties and it does not seem feasible to obtain sufficient data to establish a DRI for each [319]. Dietary recommendations for polyphenol intake should be based on bioavailability and bioactivity, taking into account inter-individual variability [269], and the DRIs should be based on from randomized placebo-controlled studies in humans [51,319].
nutrients-15-00158-t001_Table 1Table 1Effects of polyphenol-enriched supplementation on oxidative stress induced by heavy exercise training in athletes.Author and Date, SportParticipant CharacteristicsPolyphenols SupplementExercise ProtocolResultsConclusionsJówko et al., 2015 [11], sprintrandomized double-blinded study; male sprinters aged21.6 ± 1.5 years (*n* = 16); green tea extract (980 mg of PP/day taken as four capsules daily) (*n* = 8) or placebo treatment (*n* = 8) for 4 weeksdouble cycle sprint test on a bicycle ergometer with submaximal load up to 130–150 heartbeats/minsignificantly increased blood MDA, TAC and SOD in the placebo group and CK activity in both groups tested after exercise; P intake increased resting TAC levels, decreased post-exercise SOD and MDAtreatment with P prevents exercise-induced oxidative stress, but prevention of exercise-induced muscle oxidative damage or improvement in sprint performance were not identifiedJówko et al., 2012 [12], soccerrandomized double-blinded placebo-controlled design; soccer players aged 22.9 ± 5.5 years (*n* = 16) green tea PP (640 mg /day taken as two capsules) (*n* = 8) or placebo (*n* = 8) administered 2 h after breakfast muscular endurance test: three sets of two strength exercises (bench press, back press, squat) to exhaustion with a load of 60% 1RMsignificantly increased levels of TBARS, uric acid, TAS, CK and total catechins after exercise in both groupsgreen tea PP supplementation did not affect oxidative stress Morillas-Ruiz et al., 2006, [270], cycling controlled double-blinded clinical trial; male cyclists aged 23.6 ±0.9 years (*n* = 60);a beverage (blackgrape, raspberry, red currant) containing (2.3 g PP/trial) (*n* = 30) as opposed to placebo-treated*(n =* 30) submaximal 90 min aerobic exercise on a bicycle ergometer at 70% VO_2_maxno significant changesin plasma TAS levels in either group after exercise; lower increases in CK and TBARS in the supplemented group compared with the control group; decreased content of carbonyl groups in the PP-treated groupsupplementing with PP can protect against oxidative stress caused by physical exertionAllgrove et al., 2011 [308]; cyclingrandomized controlled trial in regularly exercising men aged 22 ± 4 years (*n* = 20) 40 g of dark chocolate containing 98.7 mg PP (*n* = 10), twice daily and once 2 h before training for 2 weeks, versus control group (*n* = 10)cycling for 90 min with varying VO_2_max from 60 to 90% for 30 s every 10 min, followed by this activity to exhaustion at 90% VO_2_maxsignificantly lower blood levels of F2-isoprostans during fatigue and after 1 h recovery; oxidized LDL before and after exercise; increased FFA levels in the supplemented group; no significant effect on IL-6, IL-10, IL-1Ra, glucose, glucagon, insulin and cortisol levels, and time to exhaustiondark chocolate supplementation reduced some oxidative stress markers and increased free fatty acid mobilization post-workoutSadowska-Krępa et al., 2008 [311]; swimmingmale physical education students (*n* = 14)three capsules of 390 mg/day of red grape skin extract (188 mg/g PP plus 35 mg/g anthocyanidins) (*n* = 9) three times a day for 6 weeks versus placebo (*n* = 5)interval swim test of moderate to high intensity (six repetitions of 50 m)significantly reduced CK activity, increased GSH, uric acid, TAS in plasma and increased swimming performance, minor changes in antioxidant enzymes (SOD, CAT, GPX, GR) red grape skin supplementation in sports training improved hemodynamic status and performance, and had little effect on antioxidant defense system activity Bell et al.l., 2014 [312], cyclingrandomised controlled trial; male trained cyclists aged 30 ± 8 years; (*n* = 16)30 mL of Montmorency cherry concentrate (274 mg of anthocyanins) twice daily for 7 days (*n* = 8) as opposed to a placebo (*n* = 8) a 109-min high-intensity, stochastic cycling trial on a cycle ergometer on days 5, 6, and 7significant reduction of lipid hydroperoxides in the cherry concentratelipid peroxidation was reduced with Montmorency cherry concentrate supplementationBell et al.l., 2015 [313], cyclingrandomised controlled trial; male trained cyclists aged 30 ± 8 years; (*n* = 16)30 mL of Montmorency cherry concentrate (276 mg of anthocyanins) twice daily for 8 days (*n* = 8) as opposed to a placebo (*n* = 8)a 109-min high-intensity, stochastic cycling trial on a cycle ergometer on day 5no change in plasma markers of damage (CK andlipid hydroperoxides)oxidative stress was attenuated with Montmorency cherry concentrate supplementationLevers et al., 2015, running[310]randomised controlled trial; resistance trained males aged 20.9 ± 2.6 years (*n* = 23)480 mg of powdered Montmorency cherry (*n* = 11) versus placebo (*n* = 12) × 1 per day for 10 days10 sets of 10 back squats with a barbell at 70% 1RMno change in plasma marker of oxidative damage (TBARS, MDA, TAS, SOD) compared with placebo supplementation with Montmorency cherry powder did not change markers of oxidative stress, lipid peroxidation, or antioxidant activityMcCormick et al., 2016 [314], water polodouble-blinded crossover trial; trained male water polo players (*n* = 9) aged 18.6 ±1.4 years30 mL Montmorency cherry concentrate (274 mg anthocyanins) in the morning and 60 mL of the concentrate (547 mg anthocyanins) in the evening for 6 days; no data regarding the number of participants under treatment or taking placebosimulated water polo team gameno change in plasma F2-isoprostanes between supplement and control groupssupplementing with cherry juice did not affect oxidative stressLafay et al., 2009 [266], jumpingrandomized, double-blind crossover design; elite male athletes aged 21.6 ± 2 (*n* = 9)400 mg/day of grape extract (2 capsule/day; total polyphenol content >90%; gallic acid equivalents not shown) for 1 month; no data regarding the number of participants taking a supplement or placebojumpsantioxidant capacity (ORAC) increased and no change in SOD or CAT, GPx in treated group versus control GPx decreased in the placebo grape extract maintained a healthy balance between oxidative stress and antioxidant status throughout competition periodNieman et al., 2013 [315], runningrandomized, double-blinded crossover design; male or female runners aged 19–45 years (*n* = 31)20 g blueberry polyphenol–soy protein complex (2136 mg/d gallic acid equivalents); (*n* = 16) twice a day for 17 days versus placebo(*n* = 15)2.5 h/dayrunning boutsPC did not differ between groups; no change in F2-isoprostanesno apparent benefit in the polyphenol–soy protein complex groupPP—polyphenol, PC—protein carbonyls, TAS—total antioxidant status, TAC—total antioxidant capacity, SOD—superoxide dismutase, CAT—catalase, CK—creatinine kinase, MDA—malonaldehyde, TBARS—thiobarbituric acid reactive substances, VO_2_max—maximal oxygen intake, IL-6-interleukin-6, IL-8—interleukin-8, IL-10—interleukin-10, IL-12p70—interleukin-12 antibody, IL-1Ra—interleukin-1-receptor antagonist, IL-1—interleukin-1β, TNFα—tumor necrosis factor alpha, LDH—lactate dehydrogenase, GPX—glutathione peroxidase, GR—glutathione reductase, GSH—glutathione, LDL—low density lipoproteins, FFA—free fatty acids, 1RM—one repetition maximum, HR—maximum heart rate; adapted from [290].

As described, a large body of literature has reported the antioxidative properties of dietary polyphenols in vitro, in animal models, and for non-trained participants (e.g., recreationally active, subjected to a training protocol only for the research purposes, or sedentary). These publications mostly describe the effects of mixtures of antioxidants, e.g., beverages containing polyphenols and vitamins (comprehensively reviewed in [257,259,290]). Thus, their results cannot be ascribed to polyphenols alone. It is beyond the scope of this current study to discuss data from non-athletes and concerning antioxidant supplementation strategies that are not based on dietary polyphenols alone. Unfortunately, there is limited evidence from athletes per se (i.e., professionals regularly engaged in sports) supplemented with pure polyphenols. This current review points out that the likely reasons for the conflicting results in human studies are the administration of antioxidant cocktails, limited information on the antioxidant dosage required for effectiveness, and different durations of antioxidant supplementation. Additionally, a cooperative network of cellular antioxidants is formed by interactions between exogenous antioxidants in food and endogenous antioxidants [320], which further complicates the interpretation of the research results.

A wide range of doses were used in the group of interest in this current review. Table 1 lists doses for acute and chronic supplementation with supplements containing nutritional polyphenols. For example, in [270] (Table 1), a drink with flavonoids and vitamin C was used, which does not meet the criteria for Table 2. Nonetheless, in this case the antioxidant effect was ascribed to polyphenols because the amount of vitamin C present in the beverage was the same as that found in the placebo [270]. Table 2 presents examples in athletes of acute and chronic supplementation with polyphenols alone, considering the redox-related effects only (as opposed to Table 1 that presents effects ascribable to polyphenol-enriched supplements). Contrary to expectations, discrepancies in the experimental outcomes can be seen, even when polyphenol-enriched supplements are considered (Table 1). However, this might be due to the assessment of oxidative stress using blood markers, which is by far the most commonly analyzed sampling method; these markers are not sufficient to determine the extent of oxidative stress in skeletal muscle [290]. The basic problem in the detection of free radicals lies in the fact that ROS have a short half-life and react quickly with components that control the redox state. The direct measurement of ROS levels is difficult and requires high accuracy and precision. Nonetheless, new approaches for its direct measurement are being developed [321], and procedures for less intrusive biopsies may become possible. Regarding TAS, some authors have even speculated that dietary flavonoids have a significant antioxidant effect in vivo that does not emerge from the flavonoids themselves, but from increased uric acid levels resulting from fructose metabolism after consumption of flavonoid-rich fruits [317].

Another issue is that the antioxidant activity of polyphenols appears to result from their structure [322,323]. Thus, it would be of great interest for athletes to know which classes of polyphenols are most beneficial for scavenging free radicals.

More recently, the use of polyphenols has been criticized as reviews carried out failed to demonstrate remarkable antioxidant activity in athletes [290,291]. In general, therefore, it seems that the current promotion of polyphenols as antioxidants and their supposed beneficial role in athletes’ antioxidant status is not based on unequivocal results.
nutrients-15-00158-t002_Table 2Table 2Effect of dietary polyphenol supplementation on oxidative stress induced by heavy exercise training in athletes.Author and Date, SportParticipant CharacteristicsPolyphenols SupplementExercise ProtocolResultsConclusionsDecroix et al., 2017 [324] cyclingcrossover trial; trained male cyclists aged 30 ± 3 years (*n* = 12)900 mg cocoa flavanols 1.5 h and 3 h pre-exercise (low PPdiet consumed 24 h pre-exercise); number of participants in the PP or placebo group not providedtwo cycling time-trials (each lasting 30 min at 75% of peak power output) with a passive recovery period of 100 minincreased TAC (corrected for uric acid); no change in plasma MDAacute cocoa flavanol intake had minimal effects on exercise-induced oxidative stressYarahmadi et al., 2014 [325], treadmill runningdouble-blinded clinical trial; male and female athletes (*n* = 54) aged 24.96 ± 7.37 and 22.82 ± 6.84 years100 mg supplemental anthocyanin pills; a 28-day supplementation of a single dose of 10 mL/kg/day before the run; no data regarding the number of participants under treatment or taking placebo incremental graded Bruce protocol no change in CK; no change in LDHsupplementation with anthocyanin in athletes might improve some indices of performance rather than oxidative stressMcAnulty et al., 2013 [326], runningdouble-blinded crossover trial; trained males aged 18–40 years (*n* = 14)120 mg resveratrol and 225 mg quercetin for 6 days and 240 mg resveratrol and 450 mg quercetin on day 7 immediately prior to exercise; number of participants in the PP or placebo group not provided1 h run at a 3% grade, approximately 80% VO_2_maxpost-exercise PC, FRAP, ORAC, and TEAC significantly increased but not affected by PP treatment; a significant reduction in the post-exercise F2- isoprostanes in the PP groupresveratrol and quercetin supplementation attenuates post-exercise oxidative stress by reducing F2-isoprostanesGiuriato et al. 2022 [327], cyclingsingle-blinded, crossover trial; physically active males aged 22.3 ± 3.6 years (*n* = 10)capsaicin supplements (*n* = 3); 0.957 mg capsaicin/tablet versus control fiber supplements; 2 × 390 mg of capsaicin capsules; number of participants in the PP or placebo group not provideda maximal incremental test using an ergometerredox-related effects not analyzedantioxidant potential suggested due to attenuation of the development ofperipheral fatiguePP—polyphenol; CAT—catalase, CK—creatine kinase, LDH—lactate dehydrogenase, FRAP—ferric-reducing ability of plasma (FRAP), TEAC—trolox equivalent antioxidant capacity (TEAC), MDA—malondialdehyde, ORAC—oxygen radical absorptive capacity PC—protein carbonyls, PP—polyphenol, TAC—total antioxidant capacity, TBARS—thiobarbituric acid reactive substances, SOD—superoxide dismutase.

The relationship between oxidative stress and sport is highly complex. To encourage the upregulation of endogenous antioxidant defences, ROS must be released. Indeed, ROS are generated during cellular metabolism including muscle contractions. It would therefore be very useful to know to what extent athletes experience oxidative stress, as ROS generation has been observed to rise during exercise to a level that can outpace antioxidant defenses. To our knowledge, there are no rapid tests for ROS detection or oxidative stress available on the market. There are several diagnostic methods available to detect ROS or to measure antioxidant levels, but are time-consuming and require diagnosticians. Self-conviction about undergoing oxidative stress is based on an unbalanced diet and physical effort. In such situations, many athletes reach for dietary polyphenols. The failure of some of the antioxidants reviewed here to exhibit direct and significant redox-modulating actions in vivo might reflect their lack of efficacy due to low bioavailability. Most importantly, evidence is currently scarce on outcomes related to antioxidant status and the redox-modulating effects of polyphenols alone in vivo as opposed to polyphenol-enriched supplementation.

## 4. Conclusions and Future Perspectives

It is evident that professional athletes are familiar with the concept that exercise causes oxidative stress, which needs to be prevented by exogenous doses of antioxidant polyphenols. The use of antioxidant-polyphenol-rich dietary supplements can upregulate the endogenous antioxidant defence system, which in turn might have crucial consequences for preventing excessive oxidative damage and promoting recovery. However, according to the body of literature presented here, the use of polyphenols in the diet is debatable. The variability in research findings means the determination of the antioxidative effects of dietary polyphenols remains elusive. For instance, protocols are still variable and further systematically designed studies are required to strengthen the evidence. This research should include well-constructed trials in humans.

It appears that switching dietary polyphenols’ status from nutritional supplement to medicine would change their entire management. Safety certificates, clinical trials, and further work would be required. Currently, polyphenols are considered foodstuffs and therefore do not require such restrictive regulations as medicinal substances or medicines. Thus, all unanswered issues and measures concerning the antioxidant action of dietary polyphenols in athletes should come within the remit of researchers.

## Figures and Tables

**Figure 1 nutrients-15-00158-f001:**
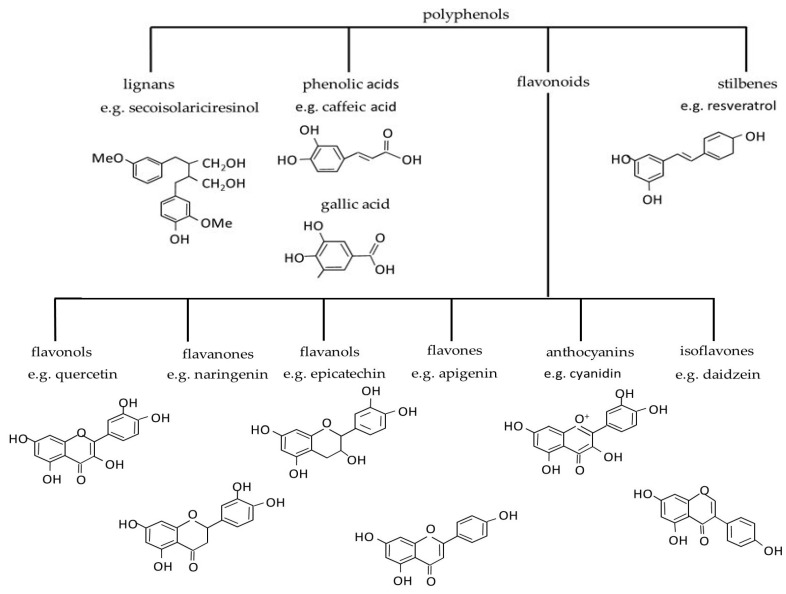
Classification and chemical structure of major classes of dietary polyphenols. Adapted from [47,48].

**Figure 2 nutrients-15-00158-f002:**
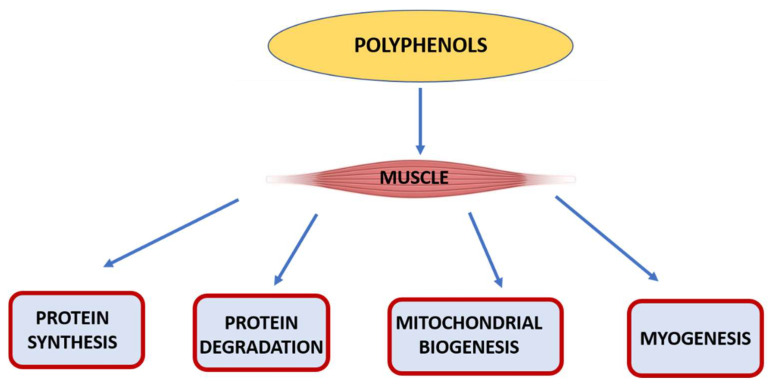
A schematic representation of the mechanism of polyphenols’ action on muscles. Adapted from [215].

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
