# Peer review of "Polyphenol Supplementation and Antioxidant Status in Athletes: A Narrative Review"

_nutrients, 2022, doi:10.3390/nu15010158_

Round 1

Reviewer 1 Report

Remarks to the article entitled:   Polyphenol supplementation and antioxidant status in athletes:  a narrative review is pretty well planned.

Before the acceptation authors should generally described the intake of phenolic compounds with  average daily diet, without supplements. After the chapter “2.2. Polyphenols bioavailability”

In mentioned above chapter authors stated that …” It is also important in establishing dietary reference intake (RDI)…. Authors should add more information why  not only dose of supplementation is important for the  RDI.  

In manuscript is many typing mistakes it should be improved for example: line 80-81, 365, 590.

Lines 212-213 please give the specific examples of polyphenols.

Line 2013-2014 the sentence is not correct.

Author Response

We thank all the reviewers for their interest in our new review on polyphenol supplementation and antioxidant status in athletes. In addition, we thank the reviewers for their thoughtful and very useful feedback below; we have amended or addressed each comment. Please read the manuscript in tracked changes.

1) Before the acceptation authors should generally described the intake of phenolic compounds with  average daily diet, without supplements. After the chapter “2.2. Polyphenols bioavailability”

 Response: We thank Reviewer#1 for these helpful comments. We have added a new section 2.3 The intake of phenolic compounds with average daily diet.

2) In mentioned above chapter authors stated that …” It is also important in establishing dietary reference intake (RDI)…. Authors should add more information why  not only dose of supplementation is important for the  RDI. 

Response We thank Reviewer#1 for these useful comments. We have addressed this in section 3.4. Polyphenols antioxidant intervention and antioxidant status in athletes. New lines have been added.

3) In manuscript is many typing mistakes it should be improved for example: line 80-81, 365, 590.

Response We thank Reviewer#1 for these useful comments. We apologize for line 80-81 where cross-referencing with Figure 1 seems to have disappeared. Thus, the following was written: “The schematic classification of polyphenols is presented in Error! Reference source not found. This is fixed now.

In line 364 the last words were “homeostasis [5])”. We apologize for not noticing this. The bracket has been deleted.

In line 365 there was “This is”. We sincerely apologize for not noticing it. These two words have been removed.

In line 590 there was: “These are summarized in Error! Reference source not found.” This is the same sort of error that occurred in lines 80-81. The cross-referencing with Table 1 disappeared. This is fixed now.

Additionally, in lines 622-624 and 626, 629-630 the same situation occurred. The cross-referencing with relevant tables was lost. This is fixed now.

4) Lines 212-213 please give the specific examples of polyphenols.

Response: We thank Reviewer#1 for these useful comments.  This has been fixed. We have added gallic and caffeic acids.

5) Line 2013-2014 the sentence is not correct. We have written “By contrast, high molecular weight polyphenols, e.g. proanthocyanidins are very poorly absorbed [113]”.

Reviewer 2 Report

I have now read the manuscript entitled: Polyphenol supplementation and antioxidant status in athletes: a narrative review.

I think the authors tried a good attempt at summarizing the effects of phenolics and polyphenols on oxidative stress and athletic performance. The subject is huge with multiple factors. I can see it is very difficult to cover it in a summarized limited narrative review. However, the authors have done their best. I have the following issues that need to be done and clarified:

1. As the authors said, stress is the important issue for athletes and their performance. well, they have not mentioned the role of stress hormone like cortisol on athletes' status.  The Enzymes, 11B-HSDs which are the gate keepers of the HPA axis, and the modulators of the active cortisol levels were not mentioned at all!

2. The sections of the review need to be stratified into the in vitro, in vivo and human studies. The review looks now some sort of a mix-up. It is difficult for the reader to check a particular issue.

3. I do not see a difference between Table 1, 2, and 3. Perhaps, a better classification of Tables would be more useful (see point 2 above).

4. It would be useful to add a figure or diagram showing the most important mechanism of action of Polyphemols on muscles.

5. There are so many errors appear in the manuscript stating that "No reference found!". Authors should resolve all these errors.

6. It would be useful if the authors can sum up their critical points on the major aspects of the review in a discussion section before the final conclusions.

Author Response

We thank all the reviewers for their interest in our new review on polyphenol supplementation and antioxidant status in athletes. In addition, we thank the reviewers for their thoughtful and very useful feedback below; we have amended or addressed each comment. Please read in tracked changes. Thank you.

  1. As the authors said, stress is the important issue for athletes and their performance. well, they have not mentioned the role of stress hormone like cortisol on athletes' status.  The Enzymes, 11B-HSDs which are the gate keepers of the HPA axis, and the modulators of the active cortisol levels were not mentioned at all!

Response We thank Reviewer#2 for these useful comments. We have added a new subheading 3.3. Physical activity, stress hormones, oxidative stress and antioxidant status.

  1. The sections of the review need to be stratified into the in vitro, in vivo and human studies. The review looks now some sort of a mix-up. It is difficult for the reader to check a particular issue.

Response We thank Reviewer#2 for these suggestions. It was not possible to stratify the manuscript because the entire manuscript would potentially be dissected into unlinked pieces.  However, we have added new information about whether specific work was done in vitro or ex vivo, etc.

  1. I do not see a difference between Table 1, 2, and 3. Perhaps, a better classification of Tables would be more useful (see point 2 above).

Response: Table 1 displays effects of phenolic compounds supplementation on oxidative stress induced by severe exercise training in humans. These data derive from a very elegant review from 2022 (Kruk et al., 2022). However, Kruk et al. presented effects of phenolic compounds supplementation on oxidative stress induced by severe exercise training in humans and exercise performance, and we removed literature positions referring to exercise performance. The most important information regarding Table 1 is that it does not contain polyphenols intake alone. For instance, in Jówko et al, 2015, they use green tea extract, which might contain antioxidant compounds, including flavonoids, epigallocatechin-3-gallate and vitamins. This means that the effects seen in Table 1 cannot be ascribed to polyphenols alone. Instead, Table 2 displays examples of acute and chronic supplementation with polyphenols alone in athletes. Table 3, similarly to Table 1 describes examples of the use polyphenol-enriched supplements. For instance, neither Montmorency chart extract might include the skin and hence, melatonin. There is credible evidence to suggest that melatonin should be classified as a mitochondria-targeted antioxidant [27–29]. None of the publications included in Table 3, apart form Levers et al. 2015, mentioned whether the skin of the Montmorency cherry was used for the preparation of the extract. In addition, none of them in the old Table 3 analyzed the extracts for melatonin. Therefore, we decided to give a title to Table 1 Effect of polyphenol-enriched supplementation on oxidative stress induced by heavy exercise training in athletes, in contrast to polyphenol-alone supplementation (Table 2). Thus, we propose to merge Table 1 and Table 3, which now will be called Table 1 Effect of polyphenol-enriched supplementation on oxidative stress induced by heavy exercise training in athletes and is based on Kruk et al., 2022 and own search.

  1. It would be useful to add a figure or diagram showing the most important mechanism of action of Polyphemols on muscles.

Response We thank Reviewer#1 for these useful comments. We have added a schematic representation of the mechanism of polyphenols' action on muscles. This is Figure 2 now. A schematic representation of the potential representative mechanisms by which polyphenols operate in the skeletal muscle. 

  1. There are so many errors appear in the manuscript stating that "No reference found!". Authors should resolve all these errors.

Response We thank Reviewer#2 for these useful comments. We apologize for line 80-81 where cross-referencing with Figure 1 seems to have disappeared. Thus, the following was written: “The schematic classification of polyphenols is presented in Error! Reference source not found. This is fixed now.

In line 364 the last words were “homeostasis [5])”. We apologize for not noticing this. The bracket has been deleted.

In line 365 there was “This is”. We sincerely apologize for not noticing it. These two words have been removed.

In line 590 there was: “These are summarized in Error! Reference source not found.” This is the same sort of error that occurred in lines 80-81. The cross-referencing with Table 1 disappeared. This is fixed now.

Additionally, in lines 622-624 and 626, 629-630 the same situation occurred. The cross-referencing with relevant tables was lost. This is fixed now.

  1. It would be useful if the authors can sum up their critical points on the major aspects of the review in a discussion section before the final conclusions.

Response: We thank Reviewer#2 for these useful comments. We have added new writing at the end of Discussion.

Round 2

Reviewer 1 Report

Short comments:

 in line 260 please provide correct calculation of years. Refeences were publish in 1976.

Reviewer 2 Report

I have now reviewed the above manuscript again entitled: Polyphenol supplementation and antioxidant status in athletes: a narrative review.

The authors have responded positively to all my comments and therefore, the manuscript has been improved to a good extent. I would like to submit my decision of Accept in the present form.